# Enantioselective synthesis of *cis*-hydrobenzofurans bearing all-carbon quaternary stereocenters and application to total synthesis of (–)-morphine

Qing Zhang[1], Fu-Min Zhang[1], Chang-Sheng Zhang[1], Si-Zhan Liu[1], Jin-Miao Tian[2], Shao-Hua Wang [1], Xiao-Ming Zhang[1] & Yong-Qiang Tu[1,2]

(–)-Morphine, which is selected as an essential medicine by World Health Organization, is widely applied in the treatment of the pain-related diseases. Due to its synthetically challenging molecular architecture and important clinical role, extensive synthetic studies of morphine-type alkaloids have been conducted. However, catalytic asymmetric total synthesis of (–)-morphine remains a long-standing challenge. Here, we disclose an efficient enantioselective total synthesis of (–)-morphine in a longest linear sequence of 16 steps. The key transformation features a highly enantioselective Robinson annulation enabled by our spiro-pyrrolidine catalyst to rapidly construct the densely functionalized *cis*-hydrodibenzofuran framework containing vicinal stereocenters with an all-carbon quaternary center. This asymmetric approach provides an alternative strategy for the synthesis of (–)-morphine and its analogues.

[1] State Key Laboratory of Applied Organic Chemistry & College of Chemistry and Chemical Engineering, Lanzhou University, Lanzhou 730000, P. R. China. [2] School of Chemistry and Chemical Engineering, Shanghai Jiao Tong University, Shanghai 200240, P. R. China. Correspondence and requests for materials should be addressed to Y.-Q.T. (email: tuyq@lzu.edu.cn) or to F.-M.Z. (email: zhangfm@lzu.edu.cn)

Morphine (**1a**), initially isolated by Sertürner from opium poppy in 1806, exhibits distinct pharmacological features, such as analgesic and sedative activities. Therefore, it is widely applied in clinical treatment and research[1]. Especially, as the most important drug used for the treatment of human pain, it was listed as an essential medicine by World Health Organization (WHO)[2]. Also its analogues were selected as the ideal medicines to cure pain-related diseases. Consequently, (−)-morphine and several related drugs have been continuously ranked among the top 200 pharmaceutical products by prescriptions according to the several statistical information by the Njardarson group[3]. Structurally, morphine has a strained pentacyclic skeleton bearing a densely functionalized *cis*-hydrodibenzofuran core and five continuous chiral centers which include a critical all-carbon quaternary stereocenter (Fig. 1). Its synthetically challenging molecular architecture and indispensable role in clinical application attract broad interests from synthetic chemists, biosynthetic chemists, and pharmaceutical chemists. Over the past 60 years, extensive synthetic studies of morphine-type alkaloids and derivatives (Fig. 1) have been conducted[4–12], resulting in both the discovery of numerous structurally related drugs to solve its undesired side effects[13,14] and the creative development of more than 30 total as well as formal syntheses based on the design of various strategies to address its potential problems of source supplies from nature. However, catalytic asymmetric total synthesis of (−)-morphine has seldom been reported[15–19], and moreover, the lack of efficient asymmetric synthetic approach is still an unsolved issue. Therefore, the exploration of an efficient asymmetric approach toward (−)-morphine is highly needed but challenging.

In the past two decades, our research interest has focused on the efficient syntheses of important bioactive natural products based on the construction of quaternary carbon centers by 1,2-carbon atom migration reactions[20,21]. In the continuation of this research interest[22–26] and further application of our spiro-pyrrolidine (SPD) organocatalysts[27–29], herein, we report an asymmetric synthetic approach toward (−)-morphine.

## Results

**Design of synthetic plan**. Synthetically, the efficient introduction of a sterically congested quaternary carbon center, especially in a catalytic fashion, is one of the main challenges in the asymmetric synthesis of (−)-morphine. Up to now, there are a few methods having been explored to achieve this purpose, for example, Pd-catalyzed Heck reaction by Overman[15], Trost[17,18] and Hudlicky[19], C–H insertion by White[16], radical cyclization by Parker[30], oxidative coupling by Gaunt[31], Grewe cyclization by Opatz[32], and Claisen rearrangement by Metz[11,33]. However, all of these approaches to the formation of the quaternary carbon require the use of chiral precursors, and catalytic enantioselective construction of the key all-carbon quaternary stereocenter from achiral substrates has not been developed. More importantly, to our best knowledge, a catalytic asymmetric approach for the assembly of AEC ring system by a direct C–C bond-forming strategy has never been reported. We envisaged our synthetic strategy could solve the above mentioned two challenges in one chemical manipulation, and the corresponding strategy was outlined in Fig. 2. The target molecule **1a** would be achieved by the cyclization of known precursor **I**, which could be obtained by the introduction of stereocenter and some functional transformations of the tetracyclic enone **II**. We envisioned the enone **II** could be synthesized by the construction of B ring from tricyclic compound **III**. Based on our recent research results on SPD-catalyzed asymmetric construction of all-carbon quaternary stereocenter[27–29], we envisaged an organocatalytic asymmetric

Robinson annulation of intermediate **IV** under suitable reaction conditions could provide this key tricyclic compound **III**. We recognized that the substrate **IV** could be generated by coupling of ethynyl amide **V** and bromide **2**, which could be prepared from commercially available compounds **3** and **4**, respectively.

**The preparation of enone precursor**. Our synthesis commenced with the preparation of compound **7** on a gram scale (Fig. 3). Starting from commercially available 3-butyn-1-ol **3**, the protection of the primary alcohol with benzyl group and subsequent treatment with *N*-methoxy-*N*-methylcarbamoyl chloride[34] afforded the Weinreb amide **5**. The boronation[35] of **5** with B$_2$(Pin)$_2$ followed by Suzuki coupling with aryl bromide **2** (prepared from 2-bromo-6-methoxyphenol **4**) furnished tri-substituted aromatic compound **6**. Final addition with methylmagnesium bromide followed by deprotection of the acetal group provided the desired precursor **7**.

**Optimization of reaction conditions**. With a sufficient amount of precursor **7** in hand, we then focused our attention on the exploration of the designed intramolecular Robinson annulation. Initially, different secondary amine catalysts were screened to catalyze the desired reaction of **7** by using benzoic acid (**A1**) as an additive in dichloromethane at room temperature (Table 1). Although no desired tricyclic product **10** was obtained, the intramolecular Michael adduct **8** was observed. Due to the lack of precedent for asymmetric synthesis of such *cis*-hydrobenzofuran skeleton bearing a sterically hindered benzylic quaternary carbon center[36–40], the current transformation would be potentially used in the syntheses of bioactive natural products and clinical drug molecules, such as abietane diterpene (−)-isoabietenin A[41] and drug (−)-galanthamine[42], which contain a common functionalized *cis*-hydrodibenzofuran nucleus (Fig. 1). More importantly, the enantio- and diastereoselectivity of the Michael reaction is vital for the access of an enantioenriched tricyclic product **10** (see Supplementary Table 5). Therefore, the investigation of the initial Michael addition was carried out. Notably, in order to accurately measure the enantiomeric excess (ee) value of the Michael adduct, the resulting aldehyde **8** was transformed in situ into its ethylester derivative **9** by Wittig reaction.

From the results listed in Table 1, the catalysts were found to have a significant influence on the enantioselectivities of the Michael reaction. Proline **Cat. 1** failed to promote this process (Table 1, entry 1), while its derivative **Cat. 2**, Jørgensen-Hayashi catalyst[43–45] **Cat. 3**, and MacMillan catalyst[46] **Cat. 4** could catalyze the desired reaction, albeit with acceptable yields and low to moderate enantioselectivities (Table 1, entries 2–4). To our delight, the SPD catalysts (**Cat. 5** and **Cat. 6**) provided the expected product **9** in good enantioselectivities along with the faster reaction rate (Table 1, entries 5–6), and the more hindered SPD catalyst **Cat. 6** gave a better result. Therefore, catalyst **Cat. 6** was selected for further optimization of other reaction parameters. By decreasing the reaction temperature to −20 °C, a higher yield and superior enantioselectivity were obtained (Table 1, entry 8). Next, various solvents and additives were screened at −20 °C (see Supplementary Tables 2–3). Fortunately, a significant improved enantioselectivity (89% ee) was observed when 2,6-dimethylbenzoic acid (**A2**) was used as the additive (Table 1, entry 9). Encouraged by this result, we thoroughly investigated other benzoic acids with diverse steric and electronic properties (see Supplementary Table 2), and found that 2,4,6-triisopropylbenzoic acid (**A3**) gave the best enantioselectivity (95% ee), while the more hindered 2,4,6-tritertbutylbenzoic acid (**A4**) had a negative effect on reaction outcomes (Table 1, entries 10–11). Finally, when the reaction temperature was lowered to −30 °C, a slight

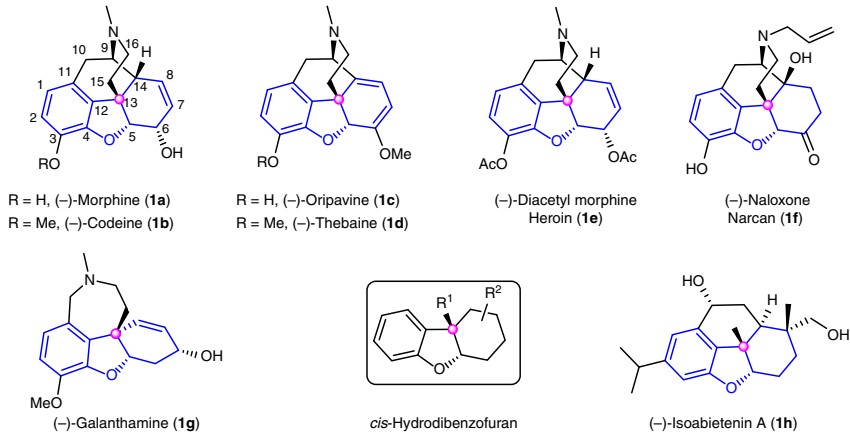

**Fig. 1** Representative natural products and derivatives containing a *cis*-hydrodibenzofuran framework. The strained *cis*-hydrodibenzofuran skeleton bearing an all-carbon quaternary stereocenter is widely found in bioactive natural products and drugs. These natural products and drugs (**1a**, **1b**, **1f**, and **1g**) have exhibited prominent biological activities and medicinal value

**Fig. 2** Our synthetic strategy towards (−)-morphine. This proposed protocol is featuring a catalytic asymmetric Robinson annulation to construct the AEC ring system of (−)-morphine with high efficiency

**Fig. 3** The preparation of compound **7** on a gram scale. The enone precursor **7** was synthesized in four steps and the key transformation included a Suzuki coupling reaction

improvement of enantioselectivity (96% ee) was observed (Table 1, entry 12). However, further decrease of temperature (−40 °C) was found to be detrimental to this reaction, resulting in a longer reaction time and no additional enhancement of enantioselectivity (Table 1, entry 13).

After establishing the optimal reaction conditions, we also explored the substrate scope of the intramolecular Michael addition (for details, see Supplementary Table 6). The reaction results demonstrated that the SPD catalyst had exhibited unique property (up to 87% yield, 96% ee, >20:1 d.r.) in this Michael addition for the construction of the challenging benzylic quaternary carbon stereocenter. More importantly, the current asymmetric transformation provides a potential platform for the preparation of a series of structurally related natural products or drugs bearing the *cis*-hydrobenzofuran skeleton.

**The enantioselective Robinson annulation.** Having successfully developed the SPD-catalyzed intramolecular Michael addition reaction, we turned our attention to the designed intramolecular Robinson annulation. Unfortunately, after screening a number of

**Table 1 The optimization of intramolecular Michael addition/Wittig reaction**

| Entry | Catalyst | Additive | T [°C] | t [h] | Yield [%][a] | d.r.[b] | ee [%][c] |
|-------|----------|----------|--------|-------|----------|-----|--------|
| 1 | 1 | A1 | rt | 72 | trace | — | — |
| 2 | 2 | A1 | rt | 72 | 60 | 3.1:1 | −34 |
| 3 | 3 | A1 | rt | 24 | 68 | 1.5:1 | 36 |
| 4 | 4 | A1 | rt | 72 | 28 | 4.1:1 | −66 |
| 5 | 5 | A1 | rt | 2 | 71 | 1.9:1 | 62 |
| 6 | 6 | A1 | rt | 1 | 73 | 3.1:1 | 72 |
| 7 | 6 | A1 | −10 | 10 | 78 | 5.2:1 | 78 |
| 8 | 6 | A1 | −20 | 24 | 82 | 7.1:1 | 80 |
| 9 | 6 | A2 | −20 | 20 | 83 | 6.9:1 | 89 |
| 10 | 6 | A3 | −20 | 20 | 86 | 7.4:1 | 95 |
| 11 | 6 | A4 | −20 | 40 | 76 | 8.7:1 | 88 |
| 12 | 6 | A3 | −30 | 48 | 87 | 10.2:1 | 96 |
| 13 | 6 | A3 | −40 | 96 | 85 | 11.8:1 | 96 |

Reaction was performed with substrate **7** (0.1 mmol), catalyst (10 mol %), and additive (20 mol %) in 1.0 mL of $CH_2Cl_2$, and followed by addition of $Ph_3P = CHCO_2Et$
[a]Isolated yield
[b]Determined by [1]H NMR prior to the addition of $Ph_3P = CHCO_2Et$
[c]Determined by chiral HPLC

catalysts including bifunctional organocatalysts, the transformation of substrate **7** to the key tricyclic product **10** proved to be ineffective, resulting in poor results (see Supplementary Table 4). Therefore, the sequent asymmetric Michael addition/aldol cyclization was next investigated[47] (Fig. 4). To our delight, when the crude Michael adduct **8** was treated with *p*-toluenesulfonic acid in toluene at 90 °C, the desired product **10** was isolated in 66% yield without the loss of enantioselectivity (94% ee). It should be noted that its enantiopurity can be further enhanced after one simple recrystallization (>99% ee, 93% yield) in ethyl acetate. Remarkably, this highly enantioselective Robinson annulation performed well even on a 5 g scale.

**Asymmetric total synthesis of (−)-codeine and (−)-morphine.** After the efficient construction of the AEC ring system of morphine, we turned our attention to assemble B ring (Fig. 5). Attempts to introduce an acetal group or its variants at α-position of enone **10** proved to be difficult. Consequently the introduction of an allyl group was next investigated. After screening various allylation reagents and reaction conditions, 3,3-dimethylallyl iodide was found to give the best results (see Supplementary Table 7), which afforded two inseparable isomers **11** (about 1:1 ratio) in 70% yield, along with an *O*-allylation by-product in 29% yield. The latter was easily transformed to the starting material **10** in 91% yield under the acidic conditions. Subsequently, the selective ozonolysis of the electron-rich double bond, followed by reduction with $Ph_3P$, led to the aldehyde, which underwent Friedel–Crafts type cyclization with a catalytic amount of

polyphosphoric acid (PPA) to afford the phenanthrofuran **12** as a single diastereoisomer. Then, the transformation of the rigid and highly strained tetracyclic enone **12** to allylic alcohol **13** was investigated. Selective epoxidation of **12** with hydrogen peroxide gave the α,β-epoxy ketone with high regio- and diastereoselectivity, which was treated with $N_2H_4 \cdot HCl$[48] to afford the allylic alcohol **13** in 54% yield over two steps. The excellent selectivity of the epoxidation on the less-hindered face and the stereospecific rearrangement of the resulting α,β-epoxy ketone produced the allylic alcohol **13** as a single isomer. This two-step procedure provides an alternative and efficient approach to transform enone to allylic alcohol derivative in comparision with the previous processes in the total synthesis of morphine[33,49]. Removal of benzyl group of **13** in the presence of electron-rich aromatic ring and disubstituted olefins proved to be a troublesome task. Debenzylation under the conventional conditions such as catalytic hydrogenolysis, Birch-type reductive cleavage, or Lewis/ Brønsted acids catalyzed cleavage was first investigated, and poor regioselectivity or complex mixtures were observed. After extensive investigations, we found that oxidative debenzylation with excess DDQ in chlorobenzene and water afforded the desired alcohol **14** in 85% yield (based on recovered starting material). The primary alcohol of compound **14** was converted to sulfonamide in preference to the allylic alcohol by a highly regioselective intermolecular Mitsunobu reaction, which was probably due to the less steric hindrance of primary alcohol. Notably, sulfonamide **15** was obtained as an enantiopure compound (>99% ee), thus suggesting no loss of enantioselectivity in the

**Fig. 4** Enantioselective Robinson annulation. The preparation of tricycle compound **10** and its X-ray crystal structure

**Fig. 5** Asymmetric total synthesis of (−)-codeine (**1b**) and (−)-morphine (**1a**). The key steps included the Friedel-Crafts reaction, Wharton reaction, Mitsunobu reaction, and hydroamination reaction

above mentioned chemical transformations. The configuration inversion of allylic alcohol **15** via a sequential one-pot oxidation-reduction process generated the Guillou's intermediate **I**[50] in high efficiency. For the crucial hydroamination reaction of **I** toward the synthesis of (−)-codeine, we developed an efficient method by using lithium 4,4′-di-*tert*-butylbiphenylide (LiDBB)[51,52] in the presence of *t*-BuOH, which efficiently afforded (−)-codeine (**1b**) in 68% yield (50 mg scale). Compared with the classical Birch-type reaction[50], photocyclization reaction[17] or oxymercuration procedure[19,53], this methodology showed a remarkable superiority in terms of reproducibility, synthetic scale, and chemical yield (see Supplementary Table 8). Finally, (−)-codeine was readily converted to (−)-morphine (**1a**) by demethylation with boron tribromide[54] in 81% yield. The spectra data of synthetic (−)-morphine and (−)-codeine are consistent with the reported ones from literatures[7,8].

## Discussion

In summary, a concise and catalytic asymmetric total synthesis of (−)-morphine was achieved in a longest linear sequence of 16 steps from commercially available 3-butyn-1-ol. The highly efficient enantioselective SPD-catalyzed Michael addition/PTSA-catalyzed cyclization sequence not only constructs the AEC tricyclic skeleton and the vicinal stereocenters of the target molecule, but also exhibits superior catalytic properties of SPD catalyst (up to 87% yield, 96% ee) in the initial Michael addition event. Moreover, the current study based on this key methodology enriches the synthetic strategy toward (−)-morphine concerning direct and catalytic asymmetric construction of the challenging all-carbon quaternary stereocenter. Additionally, except for the

Suzuki reaction, other chemical transformations can be accomplished under transition-metal-free conditions, which meets the requirement of green chemistry in drug synthesis. Meanwhile, this synthetic route also provides an alternative approach to prepare diverse derivatives of morphine for their further bioactive evaluations to address its potential clinical and social problems. Further applications of the SPD-catalyzed tandem reaction for the diverse syntheses of other natural products is ongoing in our group.

## Methods

**General information**. All moisture- or oxygen-sensitive reactions were carried out under argon atmosphere in oven-dried flasks. All solvents were purified and dried by standard techniques, and distilled prior to use. Unless otherwise noted, all reagents were analytically pure and used without further purification. All reactions were monitored by thin-layer chromatography (TLC), and the products were purified by flash column chromatography. NMR spectra were recorded in CDCl$_3$ solution on Bruker AM-400 MHz or Varian Mercury-600 MHz instruments and calibrated by using residual undeuterated solvent CHCl$_3$ (7.26 ppm) or tetra-methylsilane (0.00 ppm) as internal reference for $^1$H and the deuterated solvent CDCl$_3$ (77.00 ppm) as internal standard for $^{13}$C NMR. High-resolution mass spectra (HRMS) were measured by means of the ESI technique on Fourier transform ion cyclotron resonance mass analyzer. Chiral high performance liquid chromatography (HPLC) analysis data were recorded on a Waters e-2695 instrument equipment with Waters 2998UV/Visible detector. The X-ray single-crystal determination was performed on an Agilent SuperNova single crystal X-ray diffractometer.

**General procedure for the one-pot Michael addition/Wittig reaction**. To a stirred solution of the substrate **7** (36.8 mg, 0.1 mmol) in dry dichloromethane (1.0 mL) at −30 °C was added sequentially 2,4,6-triisopropylbenzoic acid **A3** (5.0 mg, 0.02 mmol) and **Cat. 6** (3.8 mg, 0.01 mmol). When the starting material disappeared (monitored by TLC, about 48 h), Ph$_3$P=CHCO$_2$Et was added. The reaction was warmed to room temperature and stirred for 30 min. Then the

reaction mixture was directly purified by flash column chromatography on silica gel (petroleum ether: ethyl acetate = 15:1) to give the product **9** (38 mg, 87% yield, 10.2:1 d.r., 96% ee) as a colorless oil.

## Data availability

The authors declare that the data supporting the findings of this study are available within the article and the Supplementary Information as well as from the authors upon reasonable request. The X-ray crystallographic coordinates for compound **10** reported in this study have been deposited at the Cambridge Crystallographic Data Centre (CCDC), under CCDC 1882059. These data can be obtained free of charge from The Cambridge Crystallographic Data Centre via www.ccdc.cam.ac.uk/data_request/cif. Supplementary Information and chemical compound information are available in the online version of the paper. For the proposed mechanism of the key Michael addition, see Supplementary Figure 2. For NMR analysis and HPLC traces of the compounds in this article, see Supplementary Figures 6-168.

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

## Acknowledgements

This work was financially supported by the National Natural Science Foundation of China (Nos 21772076, 21502080, 21772071, 21702136, and 21871117), National Science and Technology Major project of the Ministry of Science and Technology of China (2018ZX09711001–005–002), and the '111' Program of MOE. We also thank Prof. Chun-An Fan for his great help in the mechanism study.

## Author contributions

Q.Z. performed all of the experiments. F.-M.Z. directed the asymmetric Michael reaction and optimized the synthetic route of (-)-morphine. Q.Z., F.-M.Z., and Y.-Q.T. wrote the manuscript. C.-S.Z. prepared substrates for the synthesis of (-)-morphine. S.-Z.L. prepared substrates for reaction scope evaluation. J.-M.T. helped the design of Robinson annulation. Y.-Q.T., F.-M.Z., S.-H.W., and X.-M.Z. conceptualized and directed the project. All of the authors discussed the results and commented on the manuscript.

## Additional information

**Competing interests:** The authors declare no competing interests.

