## [Peer Review File · Nature Communications]

Reviewers' comments:

Reviewer #1 (Remarks to the Author):

This paper describes an enantioselective synthesis of morphine via an organo-catalytic Michael addition. This work is based on previous work by the same group i.e organo-catalytic Michael addition using a spiro amine as catalyst. The result described Table 2 demonstrated that the reaction provides the desired dihydrobenzofuran in good yields and high ee. The authors demonstrated the efficiency of the method by accomplishing the synthesis of (-)-morphine in 16 steps not 15 as claim by the authors. The transformation in Fig 4 is not a one pot process since the solvent (DCM) was evaporated then replaced by toluene. This is a two steps process. There are some references are not placed correctly in the text eg. "Based on our recent research results on SPD-catalyzed asymmetric construction of all-carbon quaternary stereocenter²⁷⁻²⁹" it should be ref 25 and 26. Also, the author wrote "and Claisen rearrangement by Metz¹¹" reference 11 does not correspond to any Claisen rearrangement, it should be replaced by ACIEE 2011, 3892. Overall, it is a manuscript detailing an enantioselective synthesis of dihydrobenzofuran tricycles and its applications. The fact that the synthesis is 16 steps which is not the shortest synthesis detract a bit of the application potential. I think the author would enhance the paper quality by explaining the benzoic additive effect on the e.e. and perhaps provide a scheme. I don't have any objections to accept this paper in Nat commun. after addressing the above comments.

Reviewer #2 (Remarks to the Author):

(-)-Morphine, which is selected as an essential medicine by World Health Organization, has been widely applied in the treatment of the pain-related diseases. Tu and Zhang et al. developed a catalytic asymmetric total synthesis of (-)-morphine. The key transformation features a one-pot highly enantioselective Robinson annulation enabled by they developed SPD catalyst to rapidly construct the densely functionalized cis-hydrodibenzofuran framework containing vicinal stereocenters with an all-carbon quaternary center. Compared with reported methods mediated by transition-metal, the current total synthesis represents a green synthetic strategy toward morphine by direct and catalytic asymmetric construction of the challenging benzylic quaternary carbon. So, this work is a solid piece of work, which deserves publication in Nature Communications because of its novelty and significance. For these reasons, this work is recommended for publication in Nature Communications after addressing the following issues:

1) A reference on organocatalytic asymmetric construction of chiral cyclohexenones should be cited: *Org. Biomol. Chem.*, 2014, 12, 2499–2513.

2) In the text, authors said that compound 8 could be isolated. It is better to provide its characterization data (¹H NMR, ¹³C NMR and HRMS).

3) In Table 1 and Table 2, the stereo-structure of compound 9 indicated its absolute configuration? If so, how to determine ?

4) The stereo-structure of compound 10 should be shown clearly according to the X-ray analysis (The stereo-structure of the chiral quaternary carbon should be shown in manuscript and SI).

5) In SI,

Table S3-4, yield(9) [%] and yield(9 ') [%] should be yield(10) [%] and yield(10 ') [%] ?

Table S3-5, ee (7)[%] and Δee (7-9)[%] should be ee (10)[%] and Δee (10-9)[%] ?

Pengfei Li (SUSTech)

Reviewer #3 (Remarks to the Author):

Morphine review

This manuscript describes the application of the asymmetric intramolecular Michael - aldol concept (previously demonstrated in a different ring system, ref 26) to a simple achiral molecule (7), generating a chiral tricyclic precursor (10) of the morphine system. This is indeed an efficient entry to this family of natural products.

The experiments leading to the choice of catalyst are described. Experiments showing the substrate scope of the Michael/Wittig one pot reaction should probably go in another paper.

The manuscript could be a shorter document if some of the extraneous and repetitious material were removed. Also there are some word choices that are not optimum (e.g. “ranked among” should replace “ranked as” on p 2) – in fact, some are misleading. There are a few places where the

conditional verb tense is used rather than the present tense. (On p 3, where it says “could provide,” should it not be “provides.”? Or on p 5, “could give” should be “give” if I understand what has happened. (This is a common error in translating from Chinese). In some places it is not clear that something has seldom been done or that it has never been done – e.g. p 3 “under-reported,” or p 2 “scarce.

Many structures appear more than once. This is usually unnecessary.

In summary, the novel contribution here is the asymmetric Michael reaction that gives intermediate 10. This should be focus of the paper. A major rewrite is really warranted. The current document requires too much work in order to see the essence.

Table S3-4 in the Supplementary Information has a column labeled as compound 9. I believe that should be 10. Table S3-5 has a similar problem. Clearly compound 7 has no ee. This all needs to be straightened out.

Typo p 2, sprio”

For Reviewer #1:

- 1.1)** The authors demonstrated the efficiency of the method by accomplishing the synthesis of (–)-morphine in 16 steps not 15 as claim by the authors. The transformation in Fig. 4 is not a one pot process since the solvent (DCM) was evaporated then replaced by toluene. This is a two steps process.

Response: Thank you very much for your suggestion.

We have changed “15 steps” to “16 steps” in the revised manuscript, and the corresponding “a one-pot Robinson annulation” has changed to “Robinson annulation”. According to your suggestion, we have also made several other changes in the revised manuscript.

With regard to the concept of “organocatalytic one-pot reactions”, initially, we mainly referred the reviews of Y. Hayashi and K. A. Jørgensen (Angew. Chem. Int. Ed. 2011, 50, 3605; Chem. Sci. 2016, 7, 866; Angew. Chem. Int. Ed. 2011, 50, 8492). However, according to the classical one-pot reaction and your suggestion, indeed, the transformation in Fig 4 is not a one pot process and we have also removed the sentence “Based on the effectiveness of the one-pot reaction in achieving multi-step transformation (pot economy)^{44,45},” and the corresponding references in the revised manuscript.

- 1.2)** There are some references are not placed correctly in the text eg. “Based on our recent research results on SPD-catalyzed asymmetric construction of all-carbon quaternary stereocenter²⁷⁻²⁹” it should be ref 25 and 26. Also, the author wrote “and Claisen rearrangement by Metz¹¹” reference 11 does not correspond to any Claisen rearrangement, it should be replaced by ACIE 2011, 3892.

Response: Thank you very much for your suggestion.

We have corrected abovementioned mistakes in the references part and added the literature (Angew. Chem. Int. Ed. 2011, 50, 3892) as ref. 32 in the revised manuscript. For the more details, also please see the revised manuscript.

- 1.3)** I think the author would enhance the paper quality by explaining the benzoic additive effect on the e.e. and perhaps provide a scheme.

Response: Thank you very much for your valuable suggestion.

In order to obtain the chiral product **9** with high enantioselectivity, we made great efforts toward extensive screening of the additives. Some representative experimental results have been listed in the manuscript, and the details for the effect of additives have been collected in the **Table S3-2** in the updated supplementary information.

As is shown in **Scheme R1**, firstly, the acidity of the derivatives of benzoic acid had a negligible influence on stereochemical outcome (**A1** vs **A1-1** vs **A1-2**). Subsequently, the effect on the substituents at various positions on aryl ring of benzoic acid was investigated and the results demonstrated that the substituents at C2 and C6 positions had a more significant impact on improving enantioselectivity than that of other positions (**A2** vs **A2-1** vs **A2-4**). Therefore, a

variety of substituent groups with diverse steric and electronic properties at the C2 and C6 positions were further investigated (**A2**, **A2-2**, **A2-3**, **A2-4**). In comparison with the electronic effect of substituents (**A2** vs **A2-2** vs **A2-3**), the steric hindrance of benzoic acid had an obvious effect in obtaining high enantioselectivities (**A2** vs **A2-4**). Among of them, the more sterically bulky additive **A2-4** afforded the better enantioselectivity. To further optimize the structure of additives, different substituents at C2, C4 and C6 positions of benzoic acid were also evaluated (**A2-5**, **A3**, **A4**), and finally, we found the additive **A3** gave the best result (95% ee). Unfortunately, the more hindered 2,4,6-tritertbutylbenzoic acid (**A4**) had a negative effect on enantioselectivity, maybe owing to its mismatching with the rigid catalyst **Cat. 6** in this Michael addition for the construction of quaternary carbon center.

Scheme R1. The screening of the additives

Based on the above experimental results and the well-established reactions model by aminocatalysis (*Chem. Asian. J.* 2008, 3, 922; *Acc. Chem. Res.* 2012, 45, 248), a possible stereocontrol model of this Michael addition is proposed (**Scheme R2**). Firstly, dehydration between **Cat. 6** and the aldehyde of substrate **S7** in the presence of the additive **A3** generates the iminium ion intermediate **B** (with the effect of an achiral counteranion, please see reference: *Angew. Chem. Int. Ed.* 2013, 52, 518), which can easily convert into the more stable enamine intermediate **C**. As is shown in the transition-state model **C**, both the enamine formation and a hydrogen bonding interaction between the additive **A3** and the enone moiety of **S7** probably work together to impact the stereoselectivity of the Michael addition. In this process, due to the steric hindrance of the TBDPS group in **Cat. 6**, the intramolecular nucleophilic attack from the Si face of the enone moiety is more favorable, thus leading to the formation of the quaternary center with a S configuration (if R² is C₂H₄OBn). However, presumably due to the steric hindrance of the R¹ group and R² group, the process of the other nucleophilic attack (reaction models **E** and **F**) was unfavorable. Moreover, the additive **A3** is characterized with two bulky isopropyl groups, which may effectively match with the rigid catalyst **Cat. 6** to implement a chiral reaction environment. Finally, hydrolysis of iminium ion intermediate **D** affords the product **P8** and the catalyst **Cat. 6** is released for the next catalytic cycle.

For the mechanism for stereocontrol of the Michael addition, we discussed with professor Chun-An Fan at Department of Chemistry, Lanzhou University, so the sentence “We also thank Prof. Chun-An Fan for his great help in the mechanism study.” has been added in the Acknowledgments part in the revised manuscript.

Because of the suggestion “The manuscript could be a shorter document” from reviewer 3, the proposed mechanism part has been placed in the updated supplementary information. We hope that the handling of issues about the discussion of the effect of additives and the proposed reaction model could satisfy you, editor, and reviewer 3.

Scheme R2. The proposed reaction mechanism

1.4) I don't have any objections to accept this paper in *Nat. Commun.* after addressing the above comments.

Response: Thank you very much for your valuable suggestions and comments. We hope that your suggestions and comments could be perfectly addressed and the revised manuscript and the updated supplementary information could satisfy you, editor, and other two reviewers.

For Reviewer #2:

2.1) A reference on organocatalytic asymmetric construction of chiral cyclohexenones should be cited: *Org. Biomol. Chem.*, 2014, 12, 2499–2513.

Response: *Thank you very much for your valuable suggestion.*

*We have added the literature (*Org. Biomol. Chem.*, 2014, 12, 2499.) as ref. 46 in the revised manuscript.*

2.2) In the text, authors said that compound **8** could be isolated. It is better to provide its characterization data (¹H NMR, ¹³C NMR and HRMS).

Response: *Thank you very much for your suggestion.*

*The detailed analytic data of compound **8** were listed in the updated supplementary information (Page 26).*

2.3) In Table 1 and Table 2, the stereo-structure of compound **9** indicated its absolute configuration? If so, how to determine ?

Response: *Thank you very much for your suggestion.*

*The compound **9** and compound **10** are both obtained from the common intermediate compound **8**. These chemical transformations (Wittig reaction or Aldol condensation) don't involve the change of the absolute configuration of the vicinal stereocenters. Therefore, the absolute configuration of the compound **9** can be indirectly confirmed by single crystal X-ray analysis of compound **10** (Fig. 4). The detailed single crystal X-ray analytic data of compound **10** are listed in the updated supplementary information (Page 51).*

2.4) The stereo-structure of compound **10** should be shown clearly according to the X-ray analysis (The stereo-structure of the chiral quaternary carbon should be shown in manuscript and SI).

Response: *Thank you very much for your suggestion.*

*We have offered a clearer X-ray picture of compound **10** in the revised Fig. 4 and marked out the position of the quaternary carbon (C6) of compound **10** in the updated supplementary information (Page 51).*

2.5) In SI, Table S3-4, yield (**9**) [%] and yield (**9'**) [%] should be yield (**10**) [%] and yield(**10'**) [%] ? Table S3-5, ee (**7**) [%] and Δee (**7-9**) [%] should be ee (**10**) [%] and Δee (**10-9**)[%] ?

Response: *Thank you very much for your suggestion.*

We have corrected the above mentioned mistakes in the updated supplementary information. For the more details, also please see the updated supplementary information (Table S3-4, Page 24 and Table S3-5, Page 26).

Thank you again for your helpful question and suggestion.

For Reviewer #3:

3.1) This manuscript describes the application of the asymmetric intramolecular Michael-aldol concept (previously demonstrated in a different ring system, ref 26) to a simple achiral molecule (**7**), generating a chiral tricyclic precursor (**10**) of the morphine system. This is indeed an efficient entry to this family of natural products.

***Response:** Thank you very much for your positive and valuable comments.*

*Actually, we have achieved the highly efficient and enantioselective construction of the key tricyclic skeleton (**10**) bearing a quaternary carbon center from a racemic precursor for the first time. This is an efficient approach to the syntheses of a series of structurally related natural products.*

3.2) Experiments showing the substrate scope of the Michael/Wittig one pot reaction should probably go in another paper.

***Response:** Thank you very much for your valuable suggestion.*

In our original submission, the title of the manuscript was “Catalytic asymmetric total synthesis of (–)-morphine”. Your suggestion about the removal of the substrate scope is very appropriate, and removing this part from the manuscript may be the best choice in our original version.

However, according to the suggestion from editor, the title of our revised manuscript has been changed to “Enantioselective synthesis of cis-hydrobenzofurans bearing all-carbon quaternary stereocenters and application to total synthesis of (–)-morphine”. Accordingly, in order to match the title better, the expansion of substrate scope should be carried out as an important content of this research paper.

Generally, it has been accepted in the synthetic chemistry communities that a perfect synthetic work should be a combination of the new developed methodology and its synthetic application. This revised manuscript mainly consists of two parts: the development of the asymmetric Michael addition for the construction of the synthetically challenging all-carbon quaternary centers and its application to total synthesis of important drug (–)-morphine, which are closely interrelated. As an important progress for directly constructing quaternary carbon by using our developed SPD catalyst, it is necessary to explore the substrate scope of this unprecedented intramolecular Michael addition reaction.

*Moreover, the present transformation could be used in the total syntheses of the other bioactive natural products (**Scheme R3**). In order to explore its widespread applications on the syntheses of these natural products, it is essential to investigate the various factors that affect the enantioselectivity of Michael addition, such as the steric hindrance of quaternary carbon, substitution pattern of aryl ring and electrical effect of substituent.*

And finally, in order to address your comments, both the reorganization of substrate structures in Table 2 and the discussion of substrate scope have been revised. Importantly, given the wider readers of Nature Communication from the communities of catalytic chemistry, pharmaceutical chemistry, and natural product synthetic chemistry, our current arrangement may be an ideal one. Therefore, the expansion of substrate scope has been still retained in the revised manuscript.

We do hope that you could understand our arrangement.

Scheme R3. Representative natural products

3.3) The manuscript could be a shorter document if some of the extraneous and repetitious material were removed. Many structures appear more than once. This is usually unnecessary.

Response: Thank you very much for your valuable suggestion.

According to your suggestion, many extraneous and repetitive materials (such as phrase, sentence and repetitive chemical structures) have been removed or carefully revised. Especially, both the reorganization of substrate structures in Table 2 and the corresponding description of expansion of substrate scope have been revised. Ultimately, the length of the article has reduced from 10 pages to 9 pages. For the more details, also please see the revised manuscript.

3.4) Also there are some word choices that are not optimum (e.g. “ranked among” should replace “ranked as” on p 2).

Response: Thank you very much for your valuable suggestion.

We have changed the verb phrase “ranked as” to “ranked among” (Page 2) according to your suggestion, and other mistakes which were found in manuscript have also been corrected in the revised manuscript.

3.5) There are a few places where the conditional verb tense is used rather than the present tense. (On p 3, where it says “could provide,” should it not be “provides.”? Or on p 5, “could give” should be “give” if I understand what has happened. (This is a common error in translating from Chinese).

Response: Thank you very much for your valuable suggestion.

We have changed the sentence “an organocatalytic asymmetric Robinson annulation of enone IV under suitable reaction conditions **could provide** this key tricyclic compound III.” to “**we envisaged** an organocatalytic asymmetric Robinson annulation of enone IV under suitable reaction conditions **could provide** this key tricyclic compound III.” (Page 3). Also we have changed the sentence “found that 2,4,6-triisopropylbenzoic acid (A3) **could give** the best enantioselectivity” to “found that 2,4,6-triisopropylbenzoic acid (A3) **gave** the best enantioselectivity” (Page 5) in the revised manuscript. Additional mistakes in the original manuscript have also been corrected in this revised manuscript.

3.6) In some places it is not clear that something has seldom been done or that it has never been done – e.g. p 3 “under-reported,” or p 2 “scarce”.

Response: Thank you very much for your suggestion.

According to your suggestion, we have changed “is still under-reported” to “has never been reported” (Page 2). Also we have changed “is relatively scarce” to “has seldom been reported” (Page 2) in the revised manuscript.

3.7) Table S3-4 in the Supplementary Information has a column labeled as compound **9**. I believe that should be **10**. Table S3-5 has a similar problem. Clearly compound **7** has no ee. This all needs to be straightened out.

Typo p 2, sprio”

Response: Thank you very much for your suggestion.

We have changed the compound number “**9**” to “**10**” (similarly, “**9'**” to “**10'**”) in Table S3-4. Also we changed “ee (**7**)” to “ee (**9**)” (similarly, “ee (**9**)” to “ee (**10**)”; “ee (**7-9**)” to “ee (**9-10**)”) in Table S3-5. Meanwhile, we have corrected the spelling mistake and changed “sprio” to “spiro” (Page 2). For the more details, also please see the updated supplementary information and the revised manuscript.

3.8) In summary, the novel contribution here is the asymmetric Michael reaction that gives intermediate **10**. This should be focus of the paper.

Response: Thank you very much for your valuable suggestion.

In view of the innovation of this manuscript, we quite agree with your comment. We have developed the catalytic asymmetric Michael reaction for construction of the cis-hydrodibenzofuran framework containing quaternary carbon center (**Scheme R4**).

Scheme R4. The preparation of tricyclic compound **10** using enantioselective Robinson annulation

Actually, highly efficient and catalytic asymmetric construction of such tricyclic framework (intermediate **10**) has always been a challenging topic in the past few years. Meanwhile, the compound **10** can be easily transformed from the precursor **8** by aldol condensation reaction. Thus, enantioselective synthesis of precursor **8** is also vital. Although a number of similar synthetic methods have been developed (*Org. Lett.* 2012, 14, 5526; *J. Org. Chem.* 2012, 77, 6208; *Org. Lett.* 2013, 15, 4980; *Chem. Asian J.* 2013, 8, 648; *Chem. Sci.* 2017, 8, 8086), the lack of key quaternary carbon center and the low enantioselectivities (*Org. Lett.* 2013, 15, 4980) are two major disadvantages.

In order to address this challenging theme, great efforts have been made for the investigation of the initial Michael addition. Fortunately, after extensive investigations, we developed a general and concise asymmetric approach toward this skeleton for the first time. Given the important progress, the expansion of substrate scope of the Michael addition is necessary and meaningful. Therefore, a series of substrates with various substituents have been investigated. The results demonstrated that the current transformation not only afforded the key intermediate **10** for the total synthesis of (–)-morphine, but also provided an alternative approach toward the other similar key skeleton, which could be used for synthesis of the analogues of (–)-morphine or other natural products (**Scheme R3**).

As your comment, the asymmetric preparation of intermediate **10** should be the focus of this manuscript. Therefore, based on this topic, this manuscript has been organized into the closely related two parts: asymmetric synthesis of

compound **10** and the corresponding synthetic precursor compounds, and total synthesis of (–)-morphine by using compound **10** as key intermediate, which can also match the revised title better.

Especially, as the highlight of the research paper, the novel contribution about the construction of intermediate **10** have been specifically emphasized in the Abstract and the Discussion sections in the revised manuscript. Additionally, we have also changed the subhead “One-pot enantioselective annulation” to “The preparation of tricyclic compound **10** using enantioselective Robinson annulation” in the revised manuscript.

We do hope that the corresponding response could satisfy you.

3.9) A major rewrite is really warranted. The current document requires too much work in order to see the essence.

Response: Thank you very much for your suggestion.

Above all, as mentioned in response **3.2**, the title of our original manuscript (Catalytic asymmetric total synthesis of (–)-morphine) can't match the research contents well. Moreover, there are indeed some common problems of English writing in our manuscript. For example, some word choices are not optimum or clear and some sentences are not concise enough; besides, several identical chemical structures are shown repeatedly. These issues actually affect the readers' understanding and appreciation of the original work. Therefore, the rewriting of this manuscript is really necessary.

However, according to the suggestion of editor, the title of revised manuscript has been changed to “Enantioselective synthesis of cis-hydrobenzofurans bearing all-carbon quaternary stereocenters and application to total synthesis of (–)-morphine”. In view of the above issue, correspondingly, the manuscript is organized as two closely related parts: the development of the asymmetric Michael addition for the construction of the challenging all-carbon quaternary centers and its application to the total synthesis of important drug (–)-morphine. Based on this topic, the logical structure of our previous manuscript mainly includes background and significance, retrosynthetic analysis, preparation of precursor, optimization of reaction conditions, substrate scope (the importance have been emphasized in Response to **3.2** and **3.8**) and synthetic application. Therefore, the logical structure of manuscript has not been changed.

Notably, with regard to the defects existing in the original manuscript, according to your suggestions, we have carefully revised the manuscript and made extensive changes to the manuscript. Some major revisions are listed as follow:

(1). In order to be consistent with the revised title, a brief summary about substrate scope has been added in the Abstract and the Discussion sections, so the sentence (“Additionally, an unprecedented and highly enantioselective intramolecular Michael addition has also been developed to afford a series of synthetically challenging cis-hydrobenzofuran derivatives bearing the benzylic quaternary carbon centers in excellent enantio- and diastereoselectivities (up to 96% ee and >20:1 dr) as well as good to high yields (up to 87% yield).”) has been added in the Abstract part and the sentence (“but also exhibits superior catalytic properties of SPD catalyst (up to 87% yield, 96% ee, >20:1 dr) in the initial Michael addition event. ”) has been revised in the Discussion part. Accordingly, the sentences in the Abstract and the Discussion sections have been polished.

- (2). *The major revision of the retrosynthetic analysis has been made. We removed three chemical structural formulas (the intermediate **II**, the intermediate **IV** and catalyst **Cat. 6** in the original version) in the revised Fig. 2. Consequently, the major content described in this part have been rewritten and become a shorter document.*
- (3). *We have also made large changes in the part of substrate scope. According to your suggestion, we removed repetitive chemical structural formulas (additive **A3** and catalyst **Cat. 6**) in Table 2. Moreover, in order to make the Table 2 more concise, we removed all the structural formulas of products (**9a-9l**) and the corresponding experimental results of Michael addition were displayed in more detail in the form of lists. Meanwhile, the sentences in the section have also been polished.*
- (4). *In the part of the synthetic application, we removed repetitive chemical structural formula (compound **10**) in Fig. 5. Moreover, only some key issues (important method or optimization of reaction conditions) was discussed in detail in this section and other descriptions in certain steps (such as the alkylation of **10**, the confirmation of stereochemistry of **12**, the debenylation of **13**, and the hydroamination reaction of **1**) have been removed to the updated supplementary information.*
- (5). *Finally, according to the suggestion of referee 1, we proposed a possible stereocontrol model of the Michael addition and added the discussion about the proposed reaction mechanism (Page 27) in the updated supplementary information (Page 27).*

Ultimately, after the above major revisions, the length of the article has reduced from 10 pages to 9 pages. For the more details, also please see the revised manuscript and the updated supplementary information.

We do hope that your suggestions and comments could be perfectly addressed and the revised manuscript and the updated supplementary information could satisfy you, editor, and other two reviewers.

REVIEWERS' COMMENTS:

Reviewer #1 (Remarks to the Author):

Based on the revision made, I am in favor to accept this paper for publication.

Reviewer #2 (Remarks to the Author):

All the comments have been addressed and corrections have been made accordingly. So this work is recommended for publication in Nature Communications.

Pengfei Li

Reviewer #3 (Remarks to the Author):

The authors have made a conscientious attempt to answer the earlier comments from the reviewers. However, the presentation undermines the nice chemistry.

The distraction of the “scope of the substrate” discussion remains. There’s not a lot of variety in the structure of the substrates and the generality shown is not relevant to the “representative natural products” in figure 1. The manuscript would tell a better story if Table 2 were placed in the Supplementary Material.

I don’t see that Table 2 has been changed except that its format is different. Less attractive in my opinion but, OK, do it this way.

Some errors in English usage and repeating of structures still appear (3, 4, 5 = V, 2, 7). The caption for structure 10 in the heading for Table 1 now has an extraneous “be”. Also, it is not clear why the structure “cis-hydrodibenzofuran” appears in fig 1. Check the spelling of iodide. The manuscript needs a careful reading for English and typos.

For Reviewer #1:

2.1) Based on the revision made, I am in favor to accept this paper for publication.

Response: Thank you very much for your positive comments.

For Reviewer #2:

3.1) All the comments have been addressed and corrections have been made accordingly. So this work is recommended for publication in Nature Communications.

Response: Thank you very much for your positive comments.

For Reviewer #3:

4.1) The manuscript would tell a better story if Table 2 were placed in the Supplementary Material.

Response: Thank you very much for your valuable suggestion.

According to the suggestions from you and editor, the original Table 2 and the corresponding discussion of the substrate scope have been placed in the updated supplementary information (see supplementary Table 6). For the more details, also please see the revised manuscript and the updated supplementary information.

4.2) Some errors in English usage and repeating of structures still appear (3, 4, 5 = V, 2, 7).

Response: Thank you very much for your valuable suggestion.

According to your suggestion, we have removed the repetitive chemical structural formulas (compound 2, 3 and 4) in Figure 3. However, the intermediate V (Figure 2) generally refers to a series of Weinreb amide compounds with some kind of protective group R. Initially, we screened some protective groups of intermediate V, such as Bn, PMB and MOM. And only the compound 5 with Bn group gave the best result. Therefore, the intermediate V and compound 5 has been still retained in the revised manuscript. Meanwhile, in order to display the intramolecular Michael addition more clearly and understand the reaction better, in my opinion, retaining the compound 7 in Table 1 may be the better choice. Therefore, the compound 7 in Table 1 has also been retained.

4.3) The caption for structure 10 in the heading for Table 1 now has an extraneous “be”.

Response: Thank you very much for your valuable suggestion.

We have changed the caption “not be isolated” to “not isolated” in Table 1 according to your suggestion.

4.4) Also, it is not clear why the structure “cis-hydrodibenzofuran” appears in Fig 1.

Response: Thank you very much for your valuable suggestion.

The cis-hydrodibenzofuran scaffold bearing the all-carbon quaternary stereocenter is widely found in bioactive natural products and clinical drugs (Fig. 1). In order to construct this framework efficiently, we designed and developed enantioselective Robinson annulation and achieved the total synthesis of (–)-morphine. More importantly, the key methodology would be potentially used in the syntheses of other bioactive natural products, such as (–)-isoabietenin A and (–)-galanthamine (Fig. 1). Therefore, it is useful to place the structure of “cis-hydrodibenzofuran” in Fig. 1.

In addition, in order to clearly express the above mentioned point of view and make the readers understand the importance of cis-hydrodibenzofuran better, according to your suggestion, we have emphasized the reasons for the appearance of “cis-hydrodibenzofuran” in Fig 1 in the revised manuscript. Specifically, we have make some changes in two parts. Firstly, we have revised the sentence “Structurally, morphine has a strained pentacyclic skeleton **containing a densely functionalized cis-hydrodibenzofuran core** and five continuous chiral centers which include a critical all-carbon quaternary stereocenter. (Fig. 1).” in the Introduction part. Meanwhile, the sentence “the current transformation would be potentially used in the syntheses of bioactive natural products and clinical drug molecules, such as abietane diterpene (–)-isoabietenin A and drug (–)-galanthamine, **which contain a common functionalized cis-hydrodibenzofuran nucleus (Fig. 1).**” in the part of condition optimization have also been revised. After revision, our point of view could be expressed more clearly. For the more details, also please see the revised manuscript.

4.5) Check the spelling of iodide. The manuscript needs a careful reading for English and typos.

Response: Thank you very much for your valuable suggestion.

We have changed the name of the allylation reagents from “1,1-dimethylallyl iodine” to “3,3-dimethylallyl iodide”(Page 5). According to your suggestion, we revised manuscript carefully and other mistakes which were found have also been corrected. For the more details, please see the revised manuscript.

We hope that your suggestions and comments could be perfectly addressed and the revised manuscript and the updated supplementary information could satisfy you, editor, and other two reviewers.